# How sequence alterations enhance the stability and delay expansion of DNA triplet repeat domains

Jens Völker[1] and Kenneth J. Breslauer[1,2]

[1]Department of Chemistry and Chemical Biology, Rutgers University, Piscataway, NJ, USA and [2]The Rutgers Cancer Institute of New Jersey, New Brunswick, NJ, USA

## Perspective

DNA conformational space; DNA expansion/contraction; repeat DNA origami; repeat mismatch repair; stability-functional correlations

**Corresponding author:**
Kenneth J. Breslauer;
Email: kjbdna@rutgers.edu

## Abstract

DNA sequence alterations within DNA repeat domains inexplicably enhance the stability and delay the expansion of interrupted repeat domains. Here we propose mechanisms that rationalise such unanticipated outcomes. Specifically, we describe how interruption of a DNA repeat domain restricts the ensemble space available to dynamic, slip out, repeat bulge loops by introducing energetic barriers to loop migration. We explain how such barriers arise because some possible loop isomers result in energetically costly mismatches in the duplex portion of the repeat domain. We propose that the reduced ensemble space is the causative feature for the observed delay in repeat DNA expansion. We further posit that the observed loss of the interrupting repeat in some expanded DNAs reflects the transient occupation of loop isomer positions that result in a mismatch in the duplex stem due to 'leakiness' in the energy barrier. We propose that if the lifetime of such a low probability event allows for recognition by the mismatch repair system, then 'repair' of the repeat interruption can occur; thereby rationalising the absence of the interruption in the final expanded DNA 'product.' Our proposed mechanistic pathways provide reasoned explanations for what have been described as 'puzzling' observations, while also yielding insights into a biomedically important set of coupled genotypic phenomena that map the linkage between DNA origami thermodynamics and phenotypic disease states.

### Repeat DNA sequences are inherently unstable and prone to undergo expansions, deletions, and chromosome rearrangements

Trinucleotide and higher order nucleotide repeat genomic domains are characterised by relative instability compared to more random sequence domains. This instability frequently is associated with uncontrolled DNA expansion or contraction events, as well as with chromosome rearrangements. These genomic events often correlate with debilitating phenotypic disease states (Ashley and Warren, 1995; Orr and Zoghbi, 2007). Mirkin and Khristich published a very interesting and insightful review of possible mechanisms and consequences of repeat DNA instability, entitled '*On the wrong track: molecular mechanisms of repeat mediated genome instability*' (Khristich and Mirkin, 2020). In their scholarly review, the authors critically discuss various models and conceptual frameworks that have been proposed to explain the propensity for repeats to expand or contract.

### Interruptions in pre-expanded repeat domains protect repeat DNAs against expansion while being lost in expanded repeat domains

It has been a relatively short time since DNA repeat expansion was proposed as a genotypic cause of phenotypic diseases induced by a threshold level of triplet repeat DNA expansion (Sutherland and Richards, 1995). Since then, some progress has been made towards understanding the causal relationship(s) between repeat domain instability and DNA expansion. Nonetheless, crucial features of the mechanisms of how repeat DNA instability leads to DNA expansion still remain unknown. In a subchapter within their review subtitled, '*Role of repeat interruptions in repeat stability*', Khristich and Mirkin underscore the inability of current models to account for the reduced levels of repeat expansion caused by one or more 'interruptions' within the repeat sequence domain. Employing the example of CGG repeats containing AGG interruptions, the authors also note the puzzling apparent loss of the pre-existing AGG interruption in those repeat DNA domains that have become expanded; fascinating yet perplexing observations for which meaningful explanations currently are lacking.

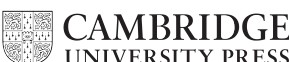

### CAA interruptions near the 3′ end of CAG repeats within the Huntingtin gene delay the onset of disease in pre-mutation length Huntington's patients

The deficiency in defining a mechanistic explanation for the origins of such interruption-induced delay in repeat expansion represents a barrier to the rational design of target-based therapeutic interventions. The need for such a mechanistic understanding is reinforced by a recent review of deep sequencing data from pre-mutation length CAG repeats in Huntington's disease patients (Wright *et al.,* 2020). In this review, Pearson and co-workers note that one or more CAA interruptions of the CAG repeats near the 3′ end of the repeat domain dramatically delay the age of onset of the Huntington's disease phenotype relative to carriers of the same length composed of uninterrupted CAG repeats. Since the CAA triplet codes for the same amino acid, glutamine, as the CAG triplet it replaces, these authors concluded that the delayed onset of Huntington's disease must be due to changes in the properties of the repeat sequence itself and not due to the polyglutamine tract in the mutated Huntingtins protein.

### Rationalising interruption-induced alterations in repeat DNA properties

Here we propose plausible explanations/mechanisms that can rationalise the impact of repeat interruptions on repeat expansion events, as well as on the puzzling apparent loss of the interrupting repeats after expansion. Our mechanistic proposals, as elaborated on below, are based on insights we have garnered from our published studies on strategically designed, CAG repeat-containing oligonucleotide systems (Völker *et al.,* 2007, 2014). Such constructs create a dynamic energy landscape shaped by nearly isoenergetic, interchanging, positional isomers that we have dubbed as 'roll-amers' (Völker *et al.,* 2008, 2012, 2019; Li *et al.,* 2014; Völker and Breslauer, 2022) (Fig. 1).

*Disruption of DNA secondary structure as a potential cause for the repeat interruption-induced increase in repeat stability.* For quite some time it has been proposed that aberrant secondary structure formation, stabilised by repeat specific intrastrand base pairing interactions, might serve as critical intermediates in the

processes that induce repeat instability leading to expansion (Gacy *et al.,* 1995; Mitas, 1997; Pearson and Sinden, 1998; McMurray, 1999; Sinden *et al.,* 2002; Lenzmeier and Freudenreich, 2003). This recognition also suggests a potential basis for the empirically observed reduction in repeat domain instability in the presence of one or more 'wrong' triplets. In this regard, it has been suggested that destabilisation of the repeat secondary structure by the 'wrong' triplet results in an unstable secondary structure, one that would not provide the same challenges to the replication and repair machinery as would be created by a 'correct' repeat secondary structure; thereby preventing the expansion process from occurring. However, as pointed out by Khristich and Mirkin in their comprehensive review (Khristich and Mirkin, 2020), it is hard to conceive how a single base change, or even a small number of repeat interruptions within very large repeat domains could manifest such an effect, and there are scant data in support of this hypothesis.

Delaney and coworkers have perhaps presented the most relevant dataset in partial support of this perspective by showing that AGG interruptions within a (CGG)n repeat oligonucleotide alter its secondary structure (as defined by susceptibility to structure-specific chemical probes), and the thermodynamic stability of the freely folding ensemble of structures adopted by (CGG)n repeat oligonucleotides (Jarem *et al.,* 2010). In this regard it should be noted that oligonucleotides composed only of repeat sequences do not fold into one unique native structure, but rather present an ensemble of interrelated folding forms, a feature that makes unambiguous interpretation of the data difficult. Given the unknown ensemble distributions in their samples, other interpretations of Delaney's results are possible. Of interest here is the interruption-induced changes in the ensemble, a feature that also plays a role, albeit in a different context, in the proposed mechanism put forward below.

### Abasic sites (and mismatches) can be accommodated within repeat DNA secondary structures without loss of secondary structure stability

In our published studies, we used abasic site lesions, inserted site specifically in place of guanine in select CAG repeats that are

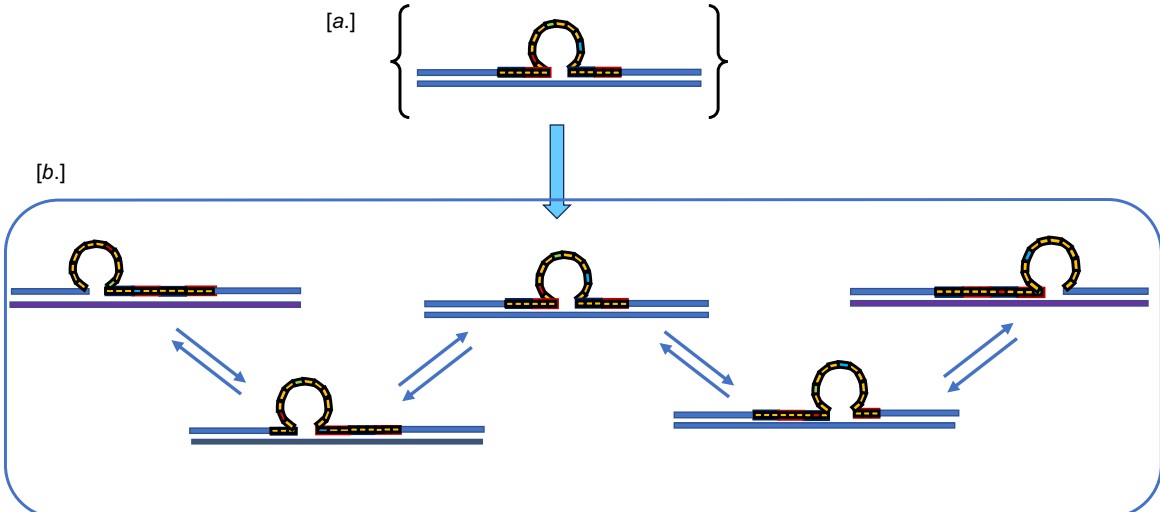

**Figure 1.** (*a*) Cartoon version of a repeat bulge loop within a larger repeat domain. Such a static representation does not highlight the reality that such repeat bulge loops really are dynamic ensembles of loop isomers as indicated in (*b*) (Völker *et al.,* 2012). Note that in these cartoon representations, we present the repeat bulge loop as a 'unstructured ring' to account for the (largely unknown) fluctuating microstructures that likely make up the repeat bulge loop ensemble within a given rollamer isomer (Völker *et al.,* 2008).

trapped within a repeat bulge loop conformation by conventional Watson and Crick base-paired domains upstream and downstream of the CAG domain (Völker *et al.,* 2009). This arrangement significantly restricts the confounding repeat ensemble distribution (Völker *et al.,* 2008), thereby simplifying the analysis. Our measurements revealed the impact of the abasic site lesion within the repeat bulge loop domain to be essentially energetically neutral. This is an unanticipated outcome since we previously have shown the abasic lesion to be one of, if not the most destabilising lesion in duplex DNA; an expectation consistent with abasic sites, in a formal sense, involving the removal of the guanine base, producing a pairing and stacking 'hole' in the DNA helix (Vesnaver *et al.,* 1989; Gelfand *et al.,* 1996, 1998; Minetti *et al.,* 2018). However, contrary to this impact on duplex DNA, we have demonstrated that the repeat self-structure is able to adjust in a compensatory manner that entirely masks the energetic cost of the loss of one putative base pair and the 5′ and 3′ associated base stacking interactions (Völker *et al.,* 2009). By contrast, introducing an 'inappropriate' base (i.e., replacing a C or G with an A to form a AGG or CAA triplet) within the repeat loop is likely to be less destabilising than an abasic site, given the plethora of non-standard, non-Watson and Crick base pairing interactions that have been shown to be only marginally less stable than the canonical Watson and Crick base pair (Nelson *et al.,* 1981; Aboul-ela *et al.,* 1985; Wu *et al.,* 1995; Santa-Lucia, 1998; SantaLucia and Hicks, 2004). Focusing on the sequence examples mentioned earlier, replacement of a CGG repeat by AGG, or CAG by CAA would result in C·A or A·C base pairs, if sequence alignment of the kind frequently proposed for repeat slip-outs is maintained. Stable DNA structures composed of A·C base pairs have been reported (Hunter *et al.,* 1986; Boulard *et al.,* 1992; Allawi and SantaLucia, 1998). Based on this analysis, it is unlikely that a single or even a few interruptions of the repeat sequence would suffice to disrupt repeat DNA self-structure to such an extent that uncontrolled expansion is delayed or absent.

Consequently, based on the body of data available, it is reasonable to conclude that one or a few interruptions to the repeat sequences are insufficient to fatally disrupt repeat secondary structure, and thereby prevent expansion. As a result, one needs to look elsewhere for possible explanations.

### *Repeat DNA secondary structures are dynamic ensembles, with the ensemble distributions dictated by differential energy levels*

We propose that the answer to this conundrum lies in the dynamic nature of repeat DNA slip outs within larger repeat domains. As we have shown, short repeat slip outs within larger repeat domains result in dynamic repeat bulge loops, which we dubbed 'rollamers.' Rollamers are dynamic, interconverting, positional isomers that can form in multiple energetically equivalent positions within the larger repeat sequence (Völker *et al.,* 2012). Such an arrangement of loop distributions is favoured by a Boltzmann entropy gain. Relatively facile interconversions between different repeat loop positions over the entire repeat sequence domain results, at equilibrium, in an ensemble distribution rather than a single repeat loop isomer, with each possible loop position being approximately equally populated. Indeed, we have postulated that the fleeting nature of such bulge loops due to relatively facile loop migration may be one of the contributing factors that cause the DNA replication and repair machineries to erroneously expand (or contract) repeats when they encounter rollameric substrates. In fact, consistent with this expectation, we have shown that loop migration can cause abasic site lesions to escape processing by the critical repair enzyme APE1

(Völker and Breslauer, 2022). The presence of an abasic site lesion in place of one of the guanines in a CAG repeat, however, alters the ensemble distributions of the rollamers, since in this construct different loop positions are no longer energetically equivalent (Völker *et al.,* 2019). Under such a circumstance, the system adjusts by altering the relative populations of loop isomer states to minimise the energetic penalty caused by the lesion.

### *How repeat domain interruptions alter the repeat DNA secondary structure ensemble distributions*

Following similar reasoning, we postulate that the presence of an interruption of the repeat sequence also will cause a change in repeat bulge loop rollamer distribution within such interrupted repeat sequences. As schematically shown in Fig. 2 and elaborated on in the figure legend, the altered triplet can either form conventional base pairs within upstream or downstream duplex domains (i.e., shown by Isomers I and II in Fig. 2), or result in the altered triplet partitioning into the repeat bulge loop domain while simultaneously causing formation of a mismatch in the duplex domain (Isomers III, IV and V in Fig. 2). Isomer III represents a special case due to partitioning of the mismatch at the 5′ loop junction, most likely resulting in an enlarged loop domain. In the former case, the energetic impact of the repeat interruption to the repeat bulge loop is indistinguishable from equivalent-sized repeat bulge loop rollamers in uninterrupted repeats. In the latter case, the energetic impact of the interruption is defined by the impact of the base mismatch and potential contributions from the loop modification. As a consequence, the different loop isomers are no longer energetically equivalent, and significant changes in the populations of different loop isomer positions can be expected. For example, the presence of an abasic site within a repeat almost completely inhibits population of those loop isomers where the abasic site partitions into the duplex domain (Völker *et al.,* 2019). A similar, but perhaps less profound, effect would be expected for the energetically less costly mismatches.

Based on our abasic site data, as discussed above (Völker *et al.,* 2019), we posit that loop isomer positions resulting in a mismatch and altered repeat loop sequence are unlikely to be populated to any significant extent due to the energetic damage such modifications cause. In other words, the interruption of the repeat sequence acts like a (possibly leaky) barrier to rollameric loop distribution. By thermodynamically discouraging population of some potential rollamer positions, the interruption of the repeat sequence causes it to behave as if equivalent to a shorter length repeat domain, particularly as far as the propensity to expand is concerned, which is exactly what has been observed empirically.

The mechanism we propose by which repeat interruptions increase repeat DNA stability represents an example of the importance of the thermodynamic impact of the final state in addition to the commonly considered initial state when one tries to assess biological outcomes.

### *Low probability, repeat bulge loop isomer states present the possibility for mismatch repair processes to result in the apparent loss of the repeat interruption*

Finally, some comments regarding the apparent loss of interrupting triplets such as AGG in an expanded domain alluded to by Khristich and Mirkin in their review (and references therein) (Khristich and Mirkin, 2020). Two possible explanations exist that can be tested by inspection of the expanded sequence. If expansion

# CAG → CAA mutation in repeat 6

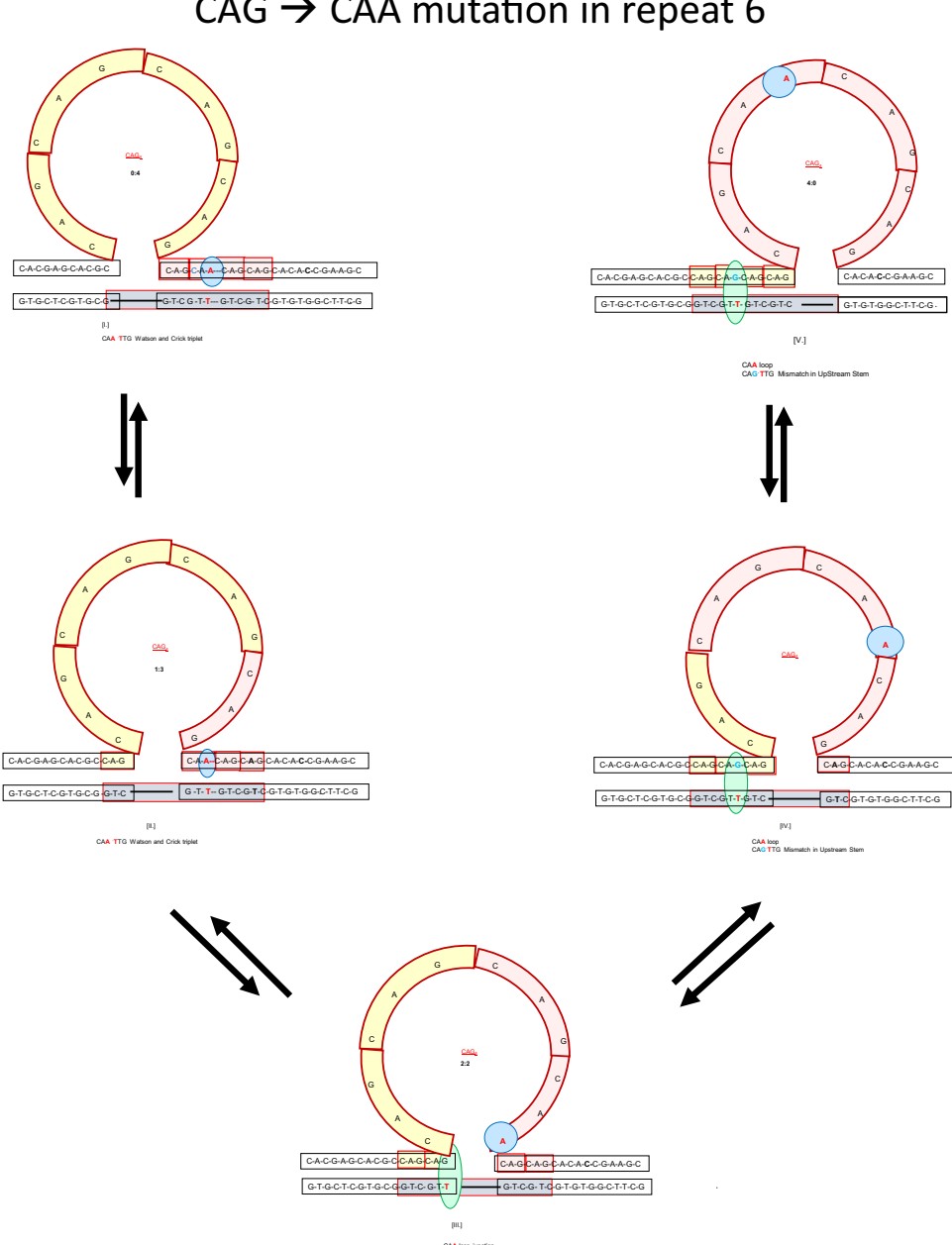

**Figure 2.** Schematic representation of the impact of repeat interruptions on the repeat bulge loop ensemble as shown for the $[CAG]_8 \cdot [CTG]_4$ system containing a CAA interruption in place of the 6th CAG repeat (Mutated base pair indicated in red letters). The $[CAG]_8 \cdot [CTG]_4$ complex results in a 4-repeat bulge loop that can be positioned in 5 possible loop positions, identified by roman numerals I-V, in the 5′ to 3′ directions. The colour coding of the CAG repeat segments into yellow (repeats 1–4) and red (repeats 5–8) is meant as a visual aid to help identify which repeat is partitioned into which domain in each of the 5 loop isomers. Loop position determines whether the CAA repeat is part of the duplex or the loop domain with the blue ball highlighting the position of the mutated A in each loop isomer. When the CAA triplet is partitioned into the loop domain, the complementary TTC triplet in the opposing strand forms base pairs with a CAG repeat that is part of the upstream duplex region, resulting in a G·T mismatch (green ellipsoid). Note that the different loop isomers can be classified into 3 general groups; as defined by the energetic impact of the repeat interruptions on the repeat bulge loop isomer; with the differential energetic impacts dictating the differential loop populations. *Group 1*: isomers I and II contain a conventionally base paired CAA/GTT triplet in the downstream duplex domain, (potential loop Isomers with the CAA/GTT triplet in the upstream duplex domain are not shown in this example but can be considered essentially equivalent); *Group 2*: Loop isomers IV and V contain a mismatched triplet CAG/GTT in the upstream duplex domain. Analogous to group 1 above, potential loop Isomers with the mismatch in the downstream duplex domain are not shown, but also can be considered essentially equivalent; and *Group 3*: Loop isomer III contains the G·T mismatch at the 5′ junction and we propose that it likely is part of an expanded loop domain. Note that a mismatch at the 3′ junction is possible, depending on the nature of the repeat interruption, as defined by the triplet sequence. An altered base in 1st, 2nd, or 3rd positions has unique impacts at the 3′ and 5′ junctions, but only impacts group 1 and 2 insofar as it alters nearest neighbours in the duplex. Fig. 2 is intended as an illustrative example. Longer repeats, larger slip outs, and repeat interruptions at different positions will produce different loop isomer arrangements, but conceptually they are represented by the three group classifications shown in Fig. 2.

happens only as a consequence of the restricted/preferred sequence space available to the 'correct' repeat bulge loop rollamer, then the expanded domain would only reflect the 'correct' repeats, while the altered triplet should still be present near the 3′ or 5′ end of the repeat domain. Alternatively, the thermodynamic argument for reduced repeat expansion propensity, due to single base/triplet interruptions of the repeat sequences, does not exclude the possibility that the repeat bulge loops adopt loop positions where their

positional partition results in the upstream duplex domain containing a mismatch, primarily corresponding to isomers IV and V in Fig. 2. Although thermodynamic considerations make population of such loop states low probability events, they would be highly consequential by resulting in repair of the interrupting repeat, hence our use of the term 'possibly leaky' barrier. This reasoning is essentially a classic thermodynamic/dynamic argument; namely, that high energy states that are sparsely populated, but do exist, can lead to additional biologically consequential processing pathways if the lifetime of the sparsely populated state is sufficiently long so as to be recognised and processed by the relevant repair enzyme. Such successful repair would remove the interrupting repeat, an outcome consistent with observation.

Given that mismatch repair has been implicated in facilitating repeat expansion events (McMurray, 2008; Iyer *et al.,* 2015; Schmidt and Pearson, 2016; Iyer and Pluciennik, 2021), this cascade of events may also trigger expansion of the now interruption-free repeat domain, also consistent with the observed outcomes from the data. The empirical observations of delayed repeat expansions in interrupted repeat sequences are a strong indicator that rollamer isomers for which loop partition result in the formation of mismatches in the repeat duplex domain are low probability events, as otherwise mismatch repair process could be expected to enhance rates of repeat expansion events.

### *Going Forward: Potential Experimental Assessments of the proposed mechanisms for interruption-induced increase in repeat DNA stability*

In closing, we wish to point out that the proposed mechanism, elaborated here, for increased stability of repeat DNA sequences with one or more interruptions, provides testable predictions that allow one to confirm or refute the proposed hypothesis. Notably, determining the thermodynamic consequences of a single interruption of the repeat sequence in various loop positions within a static repeat bulge loop will assess if the consequence of a single base disruption on repeat self-structure stability is indeed negligible, as we suggest above. Furthermore, monitoring loop distribution of a dynamic rollamer system similar to that outlined in Fig. 2, should allow one to determine if the presence of the repeat disruption indeed influences rollamer loop distribution, as hypothesised here; thereby reducing the effective repeat length. Single-molecule studies may prove useful in this regard (Hu *et al.,* 2021; Bianco *et al.,* 2022). Finally, it may be possible to test the extent to which the mismatch repair system is able to repair the repeat disruption by using a site specifically located mismatch at the repeat /nonrepeat DNA sequence junction in a rollameric system.

### *Concluding remarks*

To summarise, we have presented a novel energetic-based explanation for the puzzling observations remarked upon by Khristish and Mirkin in their review (Khristich and Mirkin, 2020) that interruptions in repeat sequences lead to increased stability and delayed expansion of these domains. Our proposed mechanism also can explain the equally puzzling observed loss of the interrupting repeat when expansion eventually does happen. Specifically, we describe for the first time how interruption of a repeat domain restricts the ensemble space available to dynamic, slip out, repeat bulge loops by introducing energetic barriers to loop migration. We present the novel proposal that these barriers arise because some possible loop isomers result in energetically costly mismatches in the duplex

portion of the repeat domain. We further propose for the first time that the reduced ensemble space is the causative feature for the observed delay in repeat DNA expansion. We further propose for the first time that the observed loss of the interrupting repeat in some expanded DNAs may be due to transient occupation of loop isomer positions that result in a mismatch in the duplex stem due to leakiness in the energy barrier. We propose the novel hypothesis that if the lifetime of such a low probability event allows for recognition by the mismatch repair system, then 'repair' of the repeat interruption can occur; thereby rationalising the absence of the interruption in the final expanded DNA 'product.'

We are pleased that Khristch and Mirkin produced such a comprehensive and provocative critical review. In the best of circumstances, such quality reviews serve as intellectual launching pads that motivate the scientific community to focus on explanations for counterintuitive observations. Their presentation of some puzzling biomedical correlative outcomes stimulated us to refocus on our biophysical studies of such systems. Our resulting proposed mechanistic pathways provide novel insights into a biomedically important set of coupled genotypic phenomena that map the linkage between DNA origami thermodynamics and phenotypic disease states.

**Open peer review.** To view the open peer review materials for this article, please visit http://doi.org/10.1017/qrd.2023.6.

**Acknowledgements.** The authors would like to thank Drs Craig A. Gelfand and G. Eric Plum for helpful discussions.

**Author contribution.** J.V. and K.J.B. contributed equally to this work.

**Competing interest.** The authors declare none.

**Financial disclosure.** Supported by grants from the NIH GM23509, GM34469, and CA47995 (all to K.J.B.).

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
