## [Reviewer Report]

*Comments to Author*: This is an interesting manuscript on an important subject, and after minor revision is certainly appropriate for publication in QRB Discovery. I would point out, however, that I found the ms as written to be very hard to read, even though I am somewhat familiar with the field of triplet expansion mechanisms, and figuring out what specifically the authors were proposing requireda careful pre-reading on my part of the excellent Khristich & Mirkin review (which is heavily cited within the present ms) before I could get any clear understanding of the specific new ideas the authors are putting forward, and how they go beyond those described in the K&M review. This may be alright, since QRB Discovery is intended as a place to publish new and developing mechanistic ideas on significant biophysical problems, and thus reviewing this complex field in detail is clearly beyond the scope of a typical Discovery paper. On the other hand, the authors could make it easier for the general reader by some reorganization of their presentation and perhaps the inclusion of a more accessible molecular mechanisms ‘cartoon’ to introduce their “rollamer” schematic (Scheme 1). Some specific suggestions follow.

1. The Abstract and the two paragraphs that follow introduce the ideas to be considered in general terms, and the section headings are helpful, but then the next paragraph on Huntington Disease seems to be dropped into the ms without any clear rationalization for what points it is supposed to make and without background. I would suggest that this paragraph be moved further back in the ms after some of the ideas it presents have been discussed in more general terms, or perhaps placed near the end to show how the authors' ideas can be applied to specific triplet expansion disease problems.

2. The injection of Scheme 1 into the ms is similarly abrupt, and is probably incomprehensible to the general reader without some preliminary introduction, possibly by drawing some sort of more familiar stick-figure cartoons used to describe triplet-expansion ideas comparable to those used by M&K in their review, to make it clearer how the new mechanistic ideas developed by the authors using their ‘rollamer’ concept fit into prior ideas of triplet expansion mechanisms.

3. It would similarly be helpful if the authors could summarize more specifically in their “Concluding Remarks” what general new concepts or ideas they have introduced into the triplet expansion field and how these ideas go beyond what is presented in the M&K review, and what particular ‘open questions’ defined by M&K their approaches help to solve. Perhaps the Huntington Disease section might fit in here to better illustrate the possibilities for using specific rollamer models approaches to provide ideas for disease therapies.

After the authors have considered the above issues and revised the ms along lines related to the suggestions made above I think this work should be quite suitable for publication in QRB Discovery, and will comprise a significant contribution.

---

## [Reviewer Report]

*Comments to Author*: In this work the authors offer an energetics-based explanation to the puzzling observation that an altered DNA sequence with a bulge expansion leads to an increase in stability and delays expansion of the affected repeat domains.I would like to emphasize the ingenuity of the offered mechanistic explanation that states that the an altered DNA sequence results in high-energy mismatches in some of the conformational states (“rollamers”) potentially available for sampling by the bulge-looped DNA molecule.Such mismatched reduce the ensemble conformational space available to the DNA while also acting act as recognition sites for DNA repair enzymes.Importantly, the authors outline experimental avenues for testing their proposed explanation.

I strongly recommend publication of this manuscript in Q. Rev. Biophys.My only comment concerns the references.Throughout the text, the references are cited mostly, but not always, in the author-date format.In the References section, however, they are presented in the numbered format.

---

## [Reviewer Report]

*Comments to Author*: Reviewer #1: In this work the authors offer an energetics-based explanation to the puzzling observation that an altered DNA sequence with a bulge expansion leads to an increase in stability and delays expansion of the affected repeat domains.I would like to emphasize the ingenuity of the offered mechanistic explanation that states that the an altered DNA sequence results in high-energy mismatches in some of the conformational states (“rollamers”) potentially available for sampling by the bulge-looped DNA molecule.Such mismatched reduce the ensemble conformational space available to the DNA while also acting act as recognition sites for DNA repair enzymes.Importantly, the authors outline experimental avenues for testing their proposed explanation.

I strongly recommend publication of this manuscript in Q. Rev. Biophys.My only comment concerns the references.Throughout the text, the references are cited mostly, but not always, in the author-date format.In the References section, however, they are presented in the numbered format.

Reviewer #3: This is an interesting manuscript on an important subject, and after minor revision is certainly appropriate for publication in QRB Discovery. I would point out, however, that I found the ms as written to be very hard to read, even though I am somewhat familiar with the field of triplet expansion mechanisms, and figuring out what specifically the authors were proposing requireda careful pre-reading on my part of the excellent Khristich & Mirkin review (which is heavily cited within the present ms) before I could get any clear understanding of the specific new ideas the authors are putting forward, and how they go beyond those described in the K&M review. This may be alright, since QRB Discovery is intended as a place to publish new and developing mechanistic ideas on significant biophysical problems, and thus reviewing this complex field in detail is clearly beyond the scope of a typical Discovery paper. On the other hand, the authors could make it easier for the general reader by some reorganization of their presentation and perhaps the inclusion of a more accessible molecular mechanisms ‘cartoon’ to introduce their “rollamer” schematic (Scheme 1). Some specific suggestions follow.

1. The Abstract and the two paragraphs that follow introduce the ideas to be considered in general terms, and the section headings are helpful, but then the next paragraph on Huntington Disease seems to be dropped into the ms without any clear rationalization for what points it is supposed to make and without background. I would suggest that this paragraph be moved further back in the ms after some of the ideas it presents have been discussed in more general terms, or perhaps placed near the end to show how the authors' ideas can be applied to specific triplet expansion disease problems.

2. The injection of Scheme 1 into the ms is similarly abrupt, and is probably incomprehensible to the general reader without some preliminary introduction, possibly by drawing some sort of more familiar stick-figure cartoons used to describe triplet-expansion ideas comparable to those used by M&K in their review, to make it clearer how the new mechanistic ideas developed by the authors using their ‘rollamer’ concept fit into prior ideas of triplet expansion mechanisms.

3. It would similarly be helpful if the authors could summarize more specifically in their “Concluding Remarks” what general new concepts or ideas they have introduced into the triplet expansion field and how these ideas go beyond what is presented in the M&K review, and what particular ‘open questions’ defined by M&K their approaches help to solve. Perhaps the Huntington Disease section might fit in here to better illustrate the possibilities for using specific rollamer models approaches to provide ideas for disease therapies.

After the authors have considered the above issues and revised the ms along lines related to the suggestions made above I think this work should be quite suitable for publication in QRB Discovery, and will comprise a significant contribution.